# Live Cell Imaging of ATP Levels Reveals Metabolic Compartmentalization within Motoneurons and Early Metabolic Changes in *FUS* ALS Motoneurons

**DOI:** 10.3390/cells12101352

**Published:** 2023-05-09

**Authors:** Vitaly L. Zimyanin, Anna-Maria Pielka, Hannes Glaß, Julia Japtok, Dajana Großmann, Melanie Martin, Andreas Deussen, Barbara Szewczyk, Chris Deppmann, Eli Zunder, Peter M. Andersen, Tobias M. Boeckers, Jared Sterneckert, Stefanie Redemann, Alexander Storch, Andreas Hermann

**Affiliations:** 1Department of Molecular Physiology and Biological Physics, School of Medicine, University of Virginia, Charlottesville, VA 22903, USA; 2Center for Membrane and Cell Physiology, School of Medicine, University of Virginia, Charlottesville, VA 22903, USA; 3Department of Neurology, Technische Universität Dresden, 01307 Dresden, Germany; julia.japtok@ukdd.de; 4Translational Neurodegeneration Section, “Albrecht Kossel”, Department of Neurology, University Medical Center Rostock, University of Rostock, 18147 Rostock, Germany; anna-maria.pielka@web.de (A.-M.P.); hannes.glass@med.uni-rostock.de (H.G.); dajana.grossman@med.uni-rostock.de (D.G.); barbara.szewczyk@med.uni-rostock.de (B.S.); 5Institute of Physiology, Technische Universität Dresden, 01307 Dresden, Germany; melanie.martin1@tu-dresden.de (M.M.); andreas.deussen@mailbox.tu-dresden.de (A.D.); 6Department of Biology, Graduate School of Arts and Sciences, University of Virginia, Charlottesville, VA 22902, USA; deppmann@virginia.edu; 7Department of Biomedical Engineering, School of Medicine, University of Virginia, Charlottesville, VA 22902, USA; ezunder@virginia.edu; 8Department of Clinical Sciences, Neurosciences, Umeå University, SE-901 85 Umeå, Sweden; peter.andersen@umu.se; 9Deutsches Zentrum für Neurodegenerative Erkrankungen (DZNE), Ulm Site, 89081 Ulm, Germany; tobias.boeckers@uni-ulm.de; 10Institute for Anatomy and Cell Biology, Ulm University, 89081 Ulm, Germany; 11Centre for Regenerative Therapie, Technische Universität Dresden, 01307 Dresden, Germany; jared.sterneckert@tu-dresden.de; 12Medical Faculty Carl Gustav Carus, Technische Universität Dresden, 01307 Dresden, Germany; 13Department of Cell Biology, School of Medicine, University of Virginia, Charlottesville, VA 22902, USA; 14Deutsches Zentrum für Neurodegenerative Erkrankungen (DZNE) Rostock/Greifswald, 18147 Rostock, Germany; 15Center for Transdisciplinary Neurosciences Rostock (CTNR), University Medical Centre, University of Rostock, 18147 Rostock, Germany; 16Department of Neurology, University of Rostock, 18147 Rostock, Germany

**Keywords:** amyotrophic lateral sclerosis, mitochondria, metabolism

## Abstract

Motoneurons are one of the most energy-demanding cell types and a primary target in Amyotrophic lateral sclerosis (ALS), a debilitating and lethal neurodegenerative disorder without currently available effective treatments. Disruption of mitochondrial ultrastructure, transport, and metabolism is a commonly reported phenotype in ALS models and can critically affect survival and the proper function of motor neurons. However, how changes in metabolic rates contribute to ALS progression is not fully understood yet. Here, we utilize hiPCS-derived motoneuron cultures and live imaging quantitative techniques to evaluate metabolic rates in fused in sarcoma (FUS)-ALS model cells. We show that differentiation and maturation of motoneurons are accompanied by an overall upregulation of mitochondrial components and a significant increase in metabolic rates that correspond to their high energy-demanding state. Detailed compartment-specific live measurements using a fluorescent ATP sensor and FLIM imaging show significantly lower levels of ATP in the somas of cells carrying FUS-ALS mutations. These changes lead to the increased vulnerability of diseased motoneurons to further metabolic challenges with mitochondrial inhibitors and could be due to the disruption of mitochondrial inner membrane integrity and an increase in its proton leakage. Furthermore, our measurements demonstrate heterogeneity between axonal and somatic compartments, with lower relative levels of ATP in axons. Our observations strongly support the hypothesis that mutated FUS impacts the metabolic states of motoneurons and makes them more susceptible to further neurodegenerative mechanisms.

## 1. Introduction

Amyotrophic lateral sclerosis (ALS) is a severe and still incurable disease that is characterized by more or less selective degeneration of upper and lower motor neurons (MN) in the motor cortex, brainstem, and spinal cord. Degeneration of motoneurons leads to the denervation of muscles and death within two to five years after symptom onset [1]. The majority of ALS cases are sporadic, with only 10% of them being monogenetically inherited. Modeling genetic ALS cases using various in vivo and in vitro systems has provided invaluable insight into the molecular mechanisms of the disease while also highlighting the inherent challenges in understanding and treating most neurodegenerative diseases [2,3,4]. These challenges are the complexity and broad scope of underlying molecular changes, the long-term and pre-symptomatic disruption of cell homeostasis that becomes critical with age and/or extra stressors, and the remarkable cell type and region specificity of neuron types vulnerable to degeneration.

Fast-fatigable motoneurons, which have the highest energy demands, are affected first and are most vulnerable in ALS while slow-firing motoneurons and sensory neurons remain relatively intact [5,6,7]. It has therefore been proposed that bioenergetic failure is one of the critical factors behind MN degeneration [8,9,10,11,12]. In support of this hypothesis, abnormal morphologies of mitochondria and disrupted function have been reported from postmortem samples of ALS patients and are an early marker in several rodent ALS models [6,11,13,14,15,16] and iPSC-derived motoneurons [10,17,18,19,20]. Changes in global metabolism and the ability to catabolize glucose by MN correlate with the severity of ALS progression [21,22]. ALS-causing mutations in *FUS* were recently reported to disrupt several energy-demanding processes in iPSC-derived MNs, such as neuronal firing rates and axonal transport [20,23,24]. Finally, FUS overexpression or mutations in various model systems disrupt mitochondrial structure and bioenergetics [13,14,15,16,25].

The exact role of FUS in regulating metabolism is far from clear. The *FUS* gene encodes a primarily nuclear protein that contains DNA/RNA binding- and low-complexity domains and plays a role in regulating transcription, DNA damage, RNA splicing, nucleocytoplasmic trafficking, ER stress, protein translation, and nonsense-mediated decay [26,27,28,29,30]. FUS binding was mapped to multiple sites on chromosomes and more than a thousand long coding mRNAs, where it was shown to regulate their splicing [31,32,33,34]. Thus, it can directly or indirectly affect multiple components involved in metabolism and its regulation. Recent examples suggest a specific enrichment of mitochondrial respiratory chain complex transcripts in FUS aggregates, resulting in the downregulation of their protein level [11]. In addition, FUS has been shown to play a very direct role in the regulation of metabolism. FUS mis-localization from the nucleus to the cytoplasm due to mutations in the nuclear-localization signal sequence or FUS overexpression results in increased interaction of FUS with enzymes involved in glucose metabolism, binding of FUS to the ATP synthase beta subunit, and induction of mitochondrial unfolded protein response in cellular and animal models [14,15].

ALS-specific mutations in FUS that change its protein-protein interactions properties or cause mis-localization to the cytoplasm and therefore an increase in the protein’s local concentration were proposed to trigger changes in the phase-transition properties of high-order protein complexes. These changes were shown to cause the maturation of FUS first into hydrogel-like states and later into solid-like irreversible fibrillar aggregates, in a so-called liquid-to-solid transition [35,36,37,38,39,40,41,42,43]. Protein aggregation is a fundamental characteristic of ALS and is a key hallmark not only of FUS and ALS but also of most other neurodegenerative diseases. The metabolic state of the cell plays a critical role in controlling such aggregate formation. The chaperone and proteasome machineries bind and neutralize misfolded proteins via refolding or degradation functions, which come at a high energy cost to the cell [44,45,46,47,48,49]. Furthermore, ATP levels in the cytoplasm have been previously shown to regulate the state of the cytoplasm and stress granules under stress conditions [50,51]. Very recent in vitro studies with ND LCDs such as FUS strongly suggest that the global metabolic state could be important in preventing aberrant assemblies of LCDs via the hydrotrope properties of ATP [38]. 

Overall, metabolism can be an essential factor contributing to ALS progression with multiple and multilayered input from the mis-regulation of ALS-causing gene targets and age-related changes in metabolism and cell homeostasis. Surprisingly, despite all of the reported phenotypes, at least one recent publication was not able to link individual FUS mutations to global changes in MN metabolism [10]. This might be due to the analysis of mixed cultures, and measurements might be masked by non-motoneuronal cells. It is possible, however, that in the absence of FUS overexpression and the additional stress often utilized in approaches to model FUS ALS, the effects of FUS mutations could be small or only occur in sub-compartments of MNs, such as the axon. The latter would indeed fit well with the mainly distal axonal trafficking deficits of mitochondria and distal axonal mitochondrial depolarization found in FUS iPSC-derived MN [23]. 

Here, we utilized human-induced pluripotent stem cell (iPSC)-derived MNs and used a range of techniques to revisit measurements of metabolic rates in motoneurons including bulk measurements as well as subcompartment-specific measurements within living neurons.

## 2. Materials and Methods

### 2.1. Cell Lines

All cell lines were obtained from skin biopsies of patients or healthy volunteers or generated using CRISP/Cas mutagenesis and have been described before (Table 1). The performed procedures were in accordance with the Declaration of Helsinki and approved by the Ethical Committee of the Technische Universität Dresden, Germany (EK 393122012 and EK 45022009) and by the Swedish Ethical Committee for Medical Research (Nr 94-135 to perform genetic research, including for FUS, and Nr 2018-494-32M to perform in vitro lab studies on cell lines derived from patients with ALS). Written informed consent was obtained from all participants, including for publication of any research results. All fibroblast lines were checked for *Mycoplasma sp.* before and after reprogramming, and afterward, routine checks for *Mycoplasma* were conducted every three to six months. We used the *Mycoplasma* detection kit according to the manufacturer’s instructions (Firma Venor GeM, No 11–1025).

### 2.2. Fibroblasts Reprogramming

Patient fibroblasts were reprogrammed using pMX-based retroviral vectors encoding the human cDNAs of OCT4, SOX2, KLF4, and cMYC (pMX vectors). Vectors were co-transfected with packaging-defective helper plasmids into 293T cells using Fugene 6 transfection reagent (Roche). Fibroblasts were plated at a density of 50,000 cells/well on 0.1% gelatin-coated 6-well plates and infected three times with a viral cocktail containing vectors expressing OCT4:SOX2:KLF4:cMYC in a 2:1:1:1 ratio in the presence of 6 µg/mL protamine sulfate (Sigma-Aldrich, St. Louis, MO, USA ) and 5 ng/mL FGF2 (Peprotech, NJ, USA). Infected fibroblasts were plated onto matrigel-coated plates at a density of 10,000 cells per 1 w/6 w in fibroblast media containing 1 mM VPA and incubated for 48 h. Afterward, the medium was changed to TeSR-E7 (Stemcell Technologies, Vancouver, BC, Canada) containing 1 mM VPA. The media was changed every day to the same conditions. VPA was added in 7–10 days. Afterward, cells were cultured in TeSR-E7 only. iPSC-like clusters started to appear at day 7 post-infection, were manually picked 14 days post-infection, and were plated onto matrigel in regular TeSR-E8 (Stemcell Technologies) medium in addition to 10 µM Y-27632 (Ascent Scientific, Bristol, UK). Stable clones were routinely passaged onto matrigel using 1 mg/ml collagenase type IV (Invitrogen, Waltham, MA, USA) and 10 µM Y-27632 (Ascent Scientific) for the first 48 h after passage. The media was changed, and fresh FGF2 was added every day.

### 2.3. NPCs and Motoneuron Generation

The generation of human NPCs and MNs was achieved by following a modified protocol from [52], as described in [23]. Colonies of iPSCs were collected, and stem cell medium was added, which contained 10 µM SB-431542, 1 µM Dorsomorphin, 3 µM CHIR 99021, and 0.5 µM pumorphamine (PMA). After 2 days, hESC medium was replaced with N2B27, consisting of the aforementioned factors and DMEM-F12/Neurobasal 50:50 supplemented with 1:200 N_2_ supplement, 1:100 B27 lacking Vitamin A, and 1% penicillin/streptomycin/glutamine. On day four, 150 µM ascorbic acid was added, while dorsomorphin and SB-431542 were withdrawn. Two days later, the EBs were mechanically separated and replated on Matrigel-coated dishes. For this purpose, Matrigel was diluted (1:100) in DMEM-F12 and kept on the dishes overnight at room temperature. Possessing a ventralized and caudalized character, the resulting so-called small molecule NPCs (smNPC) formed homogenous colonies during further cultivation. It was necessary to split them at a ratio of 1:10–1:20 once a week using Accutase for 10 min at 37 °C. Final MN differentiation was induced by treatment with 1 µM PMA in N2B27 exclusively. After 2 days, 1 µM retinoic acid (RA) was added. On day nine, another split step was performed to seed them on the desired cell culture system. Furthermore, the medium was modified to induce neural maturation. For this purpose, the developing neurons were treated with N2B27 containing 10 ng/µL BDNF, 500 µM dbcAMP, and 10 ng/µL GDNF. Following this protocol, it was possible to keep the cells in culture for up to 15–20 weeks.

### 2.4. Growth in Motoneurons in Microfluidic Chambers

The MFCs were purchased from Xona (RD900). At first, Nunc glass bottom dishes with an inner diameter of 27 mm were coated with Poly-L-Ornithine (Sigma-Aldrich P4957, 0.01% stock diluted 1:3 in PBS) overnight at 37 °C. After three steps of washing with sterile water, they were kept under the sterile hood for air drying. MFCs were sterilized with 70% ethanol and also left to dry. Next, the MFCs were dropped onto the dishes and carefully pressed on the glass surface for firm adherence. The system was then perfused with laminin (Roche 11243217001, 0.5 mg/mL stock diluted 1:50 in PBS) for 3 h at 37 °C. For seeding cells, the system was once washed with medium, and then 10 μL containing a high concentration of 3 × 10^5^ cells (3 × 10^7^ cells/mL) were directly injected into the main channel connecting two wells. After allowing for cell attachment over 30–60 min in the incubator, the still empty wells were filled up with maturation medium. This method had the advantage of increasing the density of neurons in direct juxtaposition to microchannel entries, whereas the wells remained cell-free, thereby reducing the medium turnover to a minimum. To avoid drying out, PBS was added around the MFCs. Two days after seeding, the medium was replaced in a manner that gave the neurons a guidance cue for growing through the microchannels. Specifically, a growth factor gradient was established by adding 100 μL N2B27 with 500 μM dbcAMP to the proximal seeding site and 200 μL N2B27 with 500 μM dbcAMP, 10 ng/μL BNDF, 10 ng/μL GDNF, and 100 ng/μL NGF to the distal exit site. The medium was replaced in this manner every third day. After 7 days, the first axons began spreading out at the exit site, and cells were typically maintained for up to six weeks. Mitochondrial inhibitors oligomycin A 10 μM (10 mM stock) and CCCP 10 μM (10 mM stock) were added directly into the media 1 h before imaging.

### 2.5. High-Performance Liquid Chromatography

Native AMP, ADP, and ATP were derivatized to 1,N6-etheno derivatives of endogenous adenine nucleotides as described before [54], with some modifications. First, 143 µL of the diluted samples (diluted 1:2 with bi-distilled water) were mixed with 51 µL of citrate phosphate buffer (62 and 76 mM, respectively; pH 4.0) and 6 µL chloracetaldehyde. The mixture was then incubated at 80 °C for 40 min. Derivatized samples were immediately cooled to 4 °C and stored at −20 °C until further analysis with HPLC. High-performance liquid chromatography (HPLC) was used for the quantification of the nucleotides. The HPLC system consisted of a Waters Alliance 2690 HPLC (Waters, Milford, MA, USA) coupled to a Waters 2475 fluorescence detector with an excitation wavelength of 280 nm and an emission wavelength of 410 nm. The separation was carried out on a Waters XTerra MS C18, 4.6 × 50 mm I.D. column with a particle size of 5 µm. The detailed protocol for derivatization and quantification was previously described with some modifications [53]. A guard column packed with the same sorbent was used. A total of 10 µL of sample was injected by an autosampler, and compounds were eluted with a flow rate of 0.75 mL/min using a binary tetrabutylammoniumhydrogensulfate (TBAS)/acetonitrile gradient. Eluent A contained 6% acetonitrile in the TBAS buffer, and eluent B contained 65% acetonitrile in the TBAS buffer. Initial conditions were 100% A, changed to 66% A within 5.6 min, maintained for 2.4 min, then 100% A was reestablished within 0.5 min, and the column was equilibrated for 6.5 min. External standards of known concentrations were used to check retention times and permit sample quantification based on the analysis of peak area. Empower2 (Waters) was used for the report. The ATP/ADP ratio and adenylate energy charge (AEC) have been adopted to describe the energetic status of a cell or tissue preparation. The AEC is the ratio of the competitive adenylate pool, defined as AEC = [[ATP + (0.5 ADP)]/(AMP + ADP + ATP)] [55,56,57].

### 2.6. FLIM Measurements

For fluorescence-lifetime imaging (FLIM), we used an inverted laser scanning confocal microscope (LSM 780/FLIM inverse Zeiss microscope) in the imaging facility of BIOTEC in Dresden with a temperature and CO_2_-controlled chamber. The microscope was equipped with a Becker & Hickl dual-channel FLIM unit and uses a 440 nm pulsed laser diode. For A-team imaging, we use the CFP/YFP double cube set F46-001, a 40×/1.2 LD LCI Plan-Apochromat lens, and an inverted laser scanning confocal microscope. FLIM data fitting is based on the Becker & Hickl handbook and SPCImage software (v. 5.5, Becker & Hickl). We used a two-component incomplete model to fit CFP and YFP FLIM images. The offset and scattering are set to 0. The shift is optimized to make sure the Chi2 is as close as possible to 1 (between 0.7 and 2).

### 2.7. FLIM Processing and Analysis

FLIM processing was performed using methods described in previous publications [58,59]. An important improvement was made in the normalization of photon reference images, which was carried out to compensate for varying intensities. This was accomplished using the FIJI software and the following steps: selecting Plugins -> Integral Image Filters -> Normalize Local Contrast). Next, the nucleus was zeroed, followed by cell segmentation and the creation of single-pixel ROIs using a custom ImageJ/FIJI plugin (Appendix A). The purpose of this sequence is to create pixel locations by X-Y coordinates, specific for the cytosolic cell body or axonal MN, and exclude background and noise measurements. Those locations are then applied to the FLIM data to extract any of the FLIM parameters in the data pool. Several parameters were generated, including photon images, t_1_, t_2_, a_1_%, a_2_%, and χ^2^ for each pixel of each channel, and T_m_ was calculated as T_m_ = a_1_% × t_1_ + a_2_% × t_2._ A custom-made application called Flimanalyzer (available at https://flimanalyzer.readthedocs.io/en/latest/; last version accessed: 20 February 2022) was used for the analysis of the data. The software is based on Python code and a custom-made UVA KECK center, and it performs different data combinations to produce ratios, means, and medians. The results are presented as bar graphs, frequency histograms, or normalized kernel density estimate (KDE) plots, which can be further charted using either Flimanalyzer or PRIZM. The significance tests were performed using a T-test, “two-sample assuming unequal variances,” or two-way ANOVA, with a significance level cutoff of <5% (*p* < 0.05).

### 2.8. Immunoblotting

For mitochondrial OXPHOS protein expression, proteins were extracted in RIPA buffer, and protein concentration was determined with the BCA assay (Pierce). Cell lysates were then diluted in SDS-PAGE loading buffer (Roti-Load, K929.1, Carl Roth), and SDS-PAGE was performed on 20 μg of cell lysate per sample on Tris-glycine-SDS polyacrylamide gels (4561095, Biorad), then transferred to PVDF (1620177, Biorad) using standard techniques. Primary antibodies used were against the mitochondrial respiratory chain subunits (mitochondrial OXPHOS antibody cocktail containing cytochrome c oxidase subunit 2 [COX2], cytochrome b-c1 complex subunit 2 [UQCRC2], succinate dehydrogenase [ubiquinone] flavoprotein subunit B [SDHB], NADH dehydrogenase [ubiquinone] 1 beta subcomplex subunit 8 [NDUFB8] and ATP synthase subunit alpha [ATP5A]; Cat. #ab110411, Abcam). Total protein detection (REVERT Total Protein Stain Kit, P/N 926-11010; Li-Cor) was used to normalize total cellular protein. Primary antibodies were detected with appropriate anti-mouse or anti-rabbit Infared IRDye 680 RD or 800 CW antibodies (Abcam). Protein band intensities were detected with the Odyssey 3.0 (Li-Cor Biosciences) and quantified using Image Lab 6.0 software (Biorad), and band intensity determined in the linear range was normalized to the band intensity of total protein. 

### 2.9. Isolation and Analysis of RNA and DNA

For isolation of DNA and RNA, samples were purified on days 7 of expansion, day 7 of differentiation, and days 7, 14, 21, and 28 of maturation using AllPrep DNA/RNA/Protein (80004, Qiagen, Hilden, Germany) according to the manufacturer’s instructions. To check the yield and purity of the RNA, the absorbance of the samples was measured in duplicate using a Nanodrop ND 1000 (VWR International GmbH, Darmstadt, Germany). For analysis by qPCR, RNA samples were converted to cDNA using the QuantiTect Reverse Transcription Kit (205313, Qiagen). For this purpose, the RNA samples were converted to cDNA, depending on the absorbance results, to a final concentration of 100 ng/µL, 300 ng/µL or 600 ng/µL. In the thermal cycler, samples were incubated at 42 °C for 20 min, boiled at 95 °C for 3 min, and then stopped on ice and stored at −20 °C. The qPCR was performed with the QuantiNova SYBR Green PCR Kit (208052, Qiagen). For this purpose, 10 ng of each sample was used and applied in duplicate. The primer sequences and concentrations used are shown in the table. The qPCR started with a two-minute initiation step at 95 °C. This was followed by denaturation for 5 s at 95 °C and primer hybridization with the extension of the target sequences for 10 s and 60 °C. This cycle, without the initiation step, was repeated 38 times, and a melting curve was generated at the end. The evaluation of the qPCR is performed with the Rotor-Gene Q Series software. For each primer, a concentration test was carried out before the standard curve was generated to determine the best possible threshold value through internal regression.

### 2.10. MTT Assay

Cell viability status was measured with the colorimetric MTT assay kit (ab211091, Abcam), according to the manufacturer’s protocol. Briefly, motor neuron cells were seeded into a 96-well plate (735-0465, VWR) at a density of 30 × 103 in 100 μL of medium per well in two replicates per group treatment. After 7 div or 28 div, cells were incubated for 24 h with experimental factors (oligomycin A (75351, Sigma Aldrich); CCCP (carbonyl cyanide 3-chlorophenylhydrazone) (C2759-100 MG, Sigma Aldrich); rotenone (MKBZ2534C, Sigma Aldrich); ddC (2’,3’-dideoxycytidine) (ab142240, Abcam); etoposide (Akos); 2-DG (Akos)) in standard conditions (at 37 °C in a humidified atmosphere containing 5% CO_2_). After that time, the growth medium was removed from the wells, and cells were incubated with equal volumes of growth medium and 3-(4,5-dimethylthiazol-2-yl)-2,5-diphenyltetrazolium bromide (MTT) reagent (50:50 µL) for 3 h at 37 °C. After incubation, formazan crystals were dissolved in 150 µl MTT solvent, wrapped with foil, and shaken on an orbital shaker for 15 min. Absorbance at 570 nm was read on a microplate reader (CM Sunrise™, Tecan) within one hour. Measurements within each line were normalized to the average values of mock-treated wells in each line.

### 2.11. Seahorse Measurement

Characteristic parameters of glycolysis and oxidative phosphorylation were assessed using the Seahorse XFe97 Analyzer (Agilent). Motoneurons were generated according to 2.2. NPCs and Motoneuron generation. During the split on day 9, cells were transferred to a Seahorse XF96 cell culture plate, which was PLO/Laminin coated before. Prior to Seahorse measurement cells were washed for 30 min at 37 °C with buffer-free Seahorse XF base medium. Wells on the outside of the Seahorse XF96 cell culture plate were alternatingly filled with base medium and PBS. Injection ports were loaded to block specific complexes. InjectionA—activation of glycolysis: 10 mM glucose (Sigma)/1 mM pyruvate (Thermo Fisher); InjectionB—inhibition of Complex V: 1 μM oligomycin A (Tocris); InjectionC—uncoupling: 0.7 μM FCCP (Tocris); Injection D—blocking of glycolysis and oxidative phosphorylation: 50 mM 2-DG (Sigma)/1.5 μM Rotenone (Sigma)/2.5 μM Antimycin A (Sigma). Measurement consisted of five blocks: baseline, glycolysis activation, blocking of complex V, uncoupling, blocking of glycolysis, and oxidative phosphorylation. Each block was measured in triplicate. Each measurement consisted of 10 s of mixing, followed by 10 s of settling, and 3 min of measurement. In each experiment, at least six wells per condition were measured in parallel, and only wells that showed normal run behavior (e.g., no sudden cell detachment) were analyzed. Extracellular acidification rate (ECAR) and oxygen consumption rate (OCR) were calculated individually by robust regression for each well and measurement. Means were then calculated for biological replicates (n > 7) by pooling the technical replicates (N = 6) of the same condition in each experiment.

### 2.12. NAD^+^/NADH Ratio Measurements

The experiment was performed according to the manufacturer’s description (NAD^+^/NADH Assay Kit, Abcam, ab176723). A total of 5–7 × 10^6^ cells were scraped into PBS and centrifuged for 5 min at room temperature (1500 rpm). A total of 5 × 10^6^ cells were then resuspended in 100 μL lysis buffer and incubated for 15 min at 37 °C. Subsequently, the lysates were centrifuged at 1500 rpm for 5 min. The supernatant was then used for the NAD^+^/NADH assay. A total of 25 μL NADH extraction buffer or NAD^+^ extraction buffer was added to each sample and then incubated for 15 min. For total NAD^+^/NADH measurement, samples were incubated with a 25 µL NAD^+^/NADH control solution for 15 min. Subsequently, the reaction was stopped with the opposite solution. A total of 75 μl NADH reaction mixture was added to each well, and the reaction was incubated for 30 min, protected from light. Fluorescence was measured for up to 120 min on the Spark^®^ multimode microplate reader (Tecan) at Ex/Em 540/590. Treatment with inhibitors was performed for 24 h at the following concentrations: PARG inhibitor Gallatonin (Santa Cruz, SC-202619) was used at 30 µM; PARP1 inhibitor ABT888 (Santa Cruz, SC-202901, 10 mg/mL in DMSO) was used at 2 µg/ml; and FK866 (10 mM) that induces loss of NAD^+^ by inhibiting nicotinamide phosphoribosyltransferase was used at 10 µM.

### 2.13. Lentivirus Production and Delivery

A-team and A-team R122K/R126K ORFs from original constructs [60] were sub-cloned into a pL lentiviral plasmid using Age1 and Sal1 restriction sites. Lentivirus was produced in HEK293T cells using 175 cm^2^ dishes. On the day before transfection, confluent HEK cells were trypsinized and seeded in IMDM complete media (+10% FBS) at 6–7 × 10^5^ cells/ml. For transfection, IMDM (serum-free) media and Poly(ethylenimine) (PEI) transfection reagent (Sigma) were used to deliver 10–12 µg of A-team control or A-team R122K/R126K mutant plasmids, together with pSPAX2 (6,5 µg) and pVSV-G (1–2 µg) plasmids for 30 min. After washing the transfection media, cells were then grown in complete IMDM (+10% FBS) media overnight. The next day, the media was replaced by a serum-free complete neurobasal medium with B-27 supplement and Pen/Strep (without glutamine). The virus was harvested at 24 h and 48 h time points, sterile filtered, aliquoted, and stored at −80 °C. Virus-containing MN-compatible media was added at a 1:10 ratio to freshly seeded neurons (DIV 0) and replaced with normal MN differentiation media after 24 h.

### 2.14. Mitochondrial Potential Measurements withTMRE

Motor neurons were seeded for final maturation into microfluidic chambers as described previously [23]. After 2 weeks of maturation, neurons were stained with 50 nM TMRE and 10 nM MitoTracker deep red (Molecular Probes) for 30 min at 37°. Live cell imaging was performed on a Zeiss inverted AxioObserver. Z1 microscope with LSM 900 module, using a 63× 1.4 NA plan apochromat objective and full environmental control of CO_2_, humidity, and temperature. Images were acquired from the proximal site, containing cell somata, and from the distal site, where axons sprout out from the microchannels. Images were analyzed using Fiji software. To this end, a binary mask of mitochondria was created, and the mean TMRE signal intensity was measured within the mitochondrial area. 

### 2.15. Statistics

The number of measured samples is indicated in the main text or figure legends, and for all measurements, experiments were performed on a minimum of three or more different days. Statistical analysis was performed by either a Student *t*-test or one-way ANOVA, followed by Bonferroni’s or Sidak’s multiple comparison tests. The data were analyzed using PRIZM, Excel, or Flimanalyzer software. If not mentioned otherwise, all data are displayed as means ± SEM. The significance level was set at *p* < 0.05.

## 3. Results

### 3.1. FUS Mutation Has No Effect on Motoneuronal Differentiation despite Showing Cytoplasmic FUS Mislocalization

Recent protocols for the generation of patient-specific motoneurons from iPSCs offer extremely powerful tools to model ALS in cell culture [61,62,63,64,65]. We previously generated human induced pluripotent stem cells (hiPSCs) either derived from fibroblasts of patients carrying ALS mutations or by CRISPR-based mutagenesis [23,66,67]. In this study, we also generated a new hiPSC FUS line from the fibroblasts of a patient carrying a Q23L mutation in FUS/TLS. These newly derived hiPSCs were tested for the silencing of the transforming retrovirus, expression of standard pluripotency markers (Nanog, Oct4, Sox2, and Lin28A), and ability to successfully differentiate into and stain for markers of all three germ layers (Appendix A). In our study, we used the following hiPSCs with normal karyotypes and confirmed mutations: *FUS* gene mutations in the nuclear-localization signal (NLS): P525L, R521C, and R495QsfX527, as well as a mutation in a low complexity domain (LCD) of *FUS*: Q23L (Figure 1A,B; Table 1). We have previously shown that all NLS-mutation lines derive fully functional MNs and were tested for the presence of phenotypes previously reported for ALS (Figure 1A) [23,66,68,69,70]. We could also confirm that MNs carrying mutations in the nuclear localization signal of FUS show its mis-localization to the cytoplasm (Figure 1C,D). To assess if changes in levels of FUS protein itself could be the cause of ALS-related cellular phenotypes, we compared FUS protein level changes during MN differentiation between control and lines carrying mutations in FUS NLS. All of our tested lines showed that global FUS protein cellular levels were reduced upon MN differentiation, but there was no significant difference in the amounts of this protein between control and *FUS* mutant cell lines (Figure 1E, Appendix A). 

Finally, we have revisited our previous observations of early mitochondrial defects in the distal axons of FUS-ALS MNS [23]. This time we utilized an alternative mitochondrial dye TMRE, to analyze mitochondrial membrane potential in proximal and distal axonal compartments. The analysis of the mean TMRE signal of mitochondria suggested a decrease in the mitochondrial membrane potential in FUS-P525L motoneurons compared to wild-type neurons in both the proximal and distal parts of axons (Appendix A). Furthermore, data also showed that distal mitochondria are more depolarized compared to proximal mitochondria in wild-type and FUS-P525L neurons. 

Taken together, our previously characterized protocol for an iPSC-based model of FUS-ALS showed normal differentiation into mature spinal MNs that acquire hallmark pathologies, including early visible defects in axonal mitochondria, and make a powerful model for pathophysiological studies of ALS mechanisms.

### 3.2. MN Differentiation Leads to an Increase in the Number of Mitochondrial Complex Proteins and Selected RNAs and Is Not Affected by FUS Mutations

Mature motoneurons, due to their morphology and function, are expected to have very high metabolic demands and high rates of oxidative phosphorylation (OXPHOS). It has been previously reported that MN differentiation is accompanied by a shift in their metabolic pathways from glycolysis to oxidative phosphorylation [10]. As such, we aimed to investigate whether motoneuron differentiation is associated with an overall increase in metabolic rates and OXPHOS and to re-examine the potential impact of *FUS*-ALS mutations during these processes. First, we tested if we could detect overall changes in mitochondrial material in our cell cultures during maturation. We collected cell protein extracts and mRNA samples at different stages of our motoneuron differentiation protocol: proliferating neural precursor cells (NPC), cells at the start of the differentiation protocol (pre-motoneurons, pMN), as well as MN during maturation at days 7, 14, 21, and 28 after the start of the differentiation protocol (Figure 2A). We then evaluated changes in the quantities of typical mitochondrial proteins and RNAs during differentiation. We compared three control and three independent lines carrying either P525L or R521C FUS ALS mutations (Table 1, Figure 2).

We assessed protein levels of single representative components from each of the mitochondrial respiratory chain subunits: (1) NADH dehydrogenase (ubiquinone) 1 beta subcomplex subunit 8 (NDUFB8); (2) succinate dehydrogenase (ubiquinone) flavoprotein subunit B (SDHB); (3) cytochrome b-c1 complex subunit 2 (UQCRC2); (4) cytochrome c oxidase subunit 2 (COX2); and (5) ATP synthase subunit alpha (ATP5A), as well as mitochondrial transcription factor A (mtTFA) (Figure 2). Four of the tested proteins showed a significant increase in MN by day 28 of maturation. ATP5A had a similar trend towards an increase, but the *p*-values did not reach significance (Figure 2A). 

In parallel to the above-mentioned increase in proteins of the mitochondrial complexes, we also detected a global increase in mRNA levels of Peroxisome proliferator-activated receptor gamma coactivator 1-alpha (PPARGC-1α), a transcriptional coactivator and the master regulator of mitochondrial biogenesis; Dnm1L, a mitochondria-associated, dynamin-related GTPase that mediates mitochondrial fission; and mitofusin2 (MFN-2), a mitochondrial membrane protein that participates in mitochondrial fusion and contributes to the maintenance and operation of the mitochondrial network (Figure 2C). In contrast, there was no change during MN differentiation and maturation in the expression levels of the mitochondrial dynamin-like GTPase OPA1, which regulates mitochondrial stability and energy output, or of FIS1, a component of a mitochondrial complex that promotes mitochondrial fission (Figure 2C). Of note, the presence of FUS mutations did not significantly affect levels and amounts of any of the above-tested proteins or mRNA. Thus, MN differentiation and maturation lead to an increase in several mitochondrial proteins and RNAs, consistent with the increase in their expected metabolic rates and switch to oxidative phosphorylation, which was not affected by FUS-ALS-causing mutations. 

### 3.3. NAD^+^/NADH Concentrations and Redox Ratio Increase during MN Differentiation and NAD^+^ Concentrations Increased in Mature FUS-ALS MN

Concentrations of NAD^+^ and NADH, as well as the NAD^+^ to NADH ratio, are important parameters reflecting the general metabolic and redox state of the cells [71,72,73,74]. Using cell extracts and a NAD^+^/NADH assay kit, we analyzed how NAD^+^ and NADH concentrations and their ratios are changing upon differentiation and maturation of control and FUS-ALS MN lines. Comparing averaged and individual values of all tested lines, we observed an increase in NAD^+^/NADH concentrations and redox ratios upon differentiation and maturation of MN, with the highest ratios detected in most mature MNs at day 28 (Figure 3A, Appendix A). Based on the results, it is suggested that there is an overall increase in metabolic rates and a shift toward oxidative phosphorylation in mature neuronal cells. As a positive control, we used treatment with FK866 to inhibit nicotinamide phosphoribosyltransferase, a key enzyme in NAD^+^ and NADH biosynthesis from the natural precursor nicotinamide [75]. As expected, this treatment dramatically reduced NAD^+^, and NADH levels as well as the NAD^+^/NADH ratio in all MN.

There was a significant increase in NAD^+^ concentration in mature FUS-ALS MN that carry NLS FUS mutations. However, NADH concentration or NAD^+^/NADH redox ratio values were not significantly changed between the control and FUS-ALS MN (Figure 3A, Appendix A). This could mean that mitochondrial conversion of NADH to NAD^+^ is reduced in FUS-ALS MN. 

### 3.4. Global Measurements of Adenosine Nucleotides Are Not Affected by FUS Mutations

The ATP/ADP ratio and adenylate energy charge (AEC) have been adopted to describe the energetic status of a cell or tissue preparation [55,56,76]. To further assess and compare metabolic rates and changes in FUS-ALS MNs, we decided to measure cytoplasmic levels of adenine nucleotides in the cellular lysates of mature MNs. We used 21- and 42-day-old MN cell extracts in reversed-phase high-performance liquid chromatography (RP-HPLC) as a reliable, rapid simultaneous analytical determination of ATP, ADP, and AMP concentrations (Figure 3D) [77]. We compared the ATP/ADP ratio and adenylate energy charge (AEC) between *WT* and three different FUS ALS mutant lines (R521C, R495QsfX527, and Q23L). One striking observation, consistent with our NAD^+^/NADH measurements, was that we detected a high ATP/ADP ratio in all of our MN cultures (up to a 16-fold ratio). For reference, the ratio in hepatocytes, cells known to have high metabolic rates, was shown to be 6.6 [78]. These values yet again suggest high rates of OXPHOS occurring in mature MN. However, our measurements could not detect significant differences either in ATP/ADP cytoplasmic ratio (WT = 16.25 +/− 2.29, n = 16; R521C = 17.33 +/− 0.21, n = 9, R495QsfX527 = 14.96 +/− 1.08 n = 6, Q23L = 11.68 +/− 2.3, n = 6) or in AEC (Figure 3C,D) of both day 21 and day 42 neuronal cultures carrying any of the tested FUS mutations. These data further support that FUS-ALS MNs have unchanged steady-state adenine nucleotide concentrations, which may indicate overall stable metabolic rates in our model ALS cell cultured MNs.

### 3.5. FRET-Based ATP Sensor Allows Metabolic Measurements at a Single Cell Level and Compartmental Resolution

The results of our measurements and similar published results [10] are surprising in the light of reports about mitochondrial defects, vesicle transport disruptions, and hypoexcitability observed in cell culture, in vivo models, and patient samples previously reported for FUS-ALS [13,14,15,16,25]. To overcome the potential pitfalls of heterogeneity in MN cultures and obtain live cellular-level resolution of measurements, we decided to use the FRET-based biosensor A-team to visualize ATP levels inside single living cells [60,79]. A-team1.03 employs the ε subunit of the bacterial FoF1-ATP synthase as an ATP sensory domain sandwiched between two fluorescence proteins, CFP as the donor fluorophore present N-terminally and YFP as the acceptor at the C-terminus of the ε subunit [79]. In the presence of ATP, the ε subunit binds to ATP and contorts, drawing the two fluorescence proteins closer to each other. Thus, ATP alters the fluorescent spectra of the A-team by changing the FRET efficiency between CFP and YFP. The major limitation of intensity-based FRET measurement is the assumption that all observable donor molecules undergo FRET. This is usually not the case. This varying “unbound” fraction of donor molecules introduces considerable uncertainty to the measured FRET efficiency, making comparisons between experiments impossible [80]. To overcome this problem, we used fluorescence lifetime imaging (FLIM) [80,81,82]. The lifetime is determined by building up a histogram of detected fluorescence events. This most commonly reveals a multi-exponential fluorescence decay (Figure 4C). Numerical curve fitting renders the fluorescence lifetime and the amplitude (i.e., the number of detected photons). Since FRET decreases the donor lifetime, one can quantify the extent to which FRET occurs, provided the donor lifetime without FRET is known. This donor lifetime *Tm* serves as an absolute reference against which the FRET sample is analyzed. Therefore, FLIM-FRET is internally calibrated, alleviating many of the shortcomings of intensity-based FRET measurements. As our reference donor lifetime, we used the lifetime of the CFP donor in a R122K/R126K mutant isoform of ATeam1.03 that has no detectable ATP binding and therefore no ATP-dependent FRET events but is otherwise identical to the normal sensor [60,79]. For delivery of both ATeam1.03 and ATeam1.03 R122K/R126K, we prepared a lentivirus and infected differentiating neurons with the last final re-plating of pMN, which were then matured for a further 21 days. Lower levels of virus load during infection allowed small numbers of neurons to be infected, thus enabling easy identification of individual somato-dendritic and distal axonal compartments (Figure 4A). 

As expected, the mean lifetime (T_m_) of the CFP-donor of the mutant non-FRET construct was significantly higher (i.e., low FRET levels) than that of the original functional A-team in two different *WT* control lines (wt^1^ and wt^2^) (Tm(ATeam1.03 R122K/R126K) = 1745 +/− 26.9, n = 36 versus T_m_ wt^.1^ = 1283 +/− 32.05, n = 36 and T_m_ wt^2^ = 1284 +/− 30.05, n = 24) (Figure 4B,D, Appendix A). To make sure that these lifetime changes are indeed due to decreased ATP-dependent changes in FRET of the A-team sensor, we applied two commonly used and well-characterized inhibitors of the final mitochondrial steps of OXPHOS, namely CCCP and oligomycin A, to the MN. Both inhibitors caused a very strong increase in T_m_ in treated neurons: T_m_
^CCCP^ =1581 +/− 34.43, n = 26, and T_m_Oligom =1573 +/− 36.71, n = 13), an approximately 65% relative reduction in FRET efficiency in the case of CCCP treatment (Figure 4B–D). Thus, the A-team sensor performs well in living neurons and detects a strong reduction in ATP levels upon mitochondrial inhibition of neuronal cells.

### 3.6. ATP Levels in Distal Axons Are Significantly Lower in Comparison to Soma

ALS is often reported as axonopathy since the first symptomatic changes are associated with disruption of distal axons and neuromuscular junction (NMJ) muscle denervation [13,83,84]. 

Therefore, we wanted to measure the inherent differences between ATP levels in different sub-compartments of motoneurons to establish if distal axons are different for soma. We compared average sensor lifetimes in the soma and axons of two different patient-derived cell lines from healthy individuals (Figure 5A,C). In both control lines, FLIM lifetime values (T_m_) were significantly higher in the distal axons in comparison to the soma: T_m_
^soma^ *wt^1^* =1283 +/− 32.05, n = 36 versus T_m_^axons^ *wt^.1^* =1530 +/− 44.99, n = 39, *p* = 0.0001; T_m_
^soma^ *wt^2^* = 1284 +/− 30.05, n = 27 versus T_m_^axons^ *wt^2^* = 1517 +/− 66.23, n = 13, *p* = 0.0003). These values suggest a strong relative reduction in ATP levels in the distal axon and argue that distal axons have a significant disadvantage in energy supply. Analysis of the Tm value distribution via frequency and sample size normalized kernel-density distributions (KDE) plot showed a clear shift towards lower FRET values (Appendix A). Therefore, it is unlikely to be due to a potentially smaller sample area of axonal cytoplasm. Distribution curves also suggest a clear split between values with lower and higher T_m_, suggesting a less homogenous distribution of ATP in all conditions.

### 3.7. ATP Levels across Neuron Drop during Prolonged Growth in Culture

Slowing down metabolism is a common hallmark of aging and is considered one of the critical risk factors that tip cellular homeostasis toward the symptomatic onset of ALS and neurodegenerative diseases in general [70,85]. Although this slowing down of metabolism is a common assumption, precise measurements in specific cells have not been conducted extensively in vivo, especially in specific subsets of neuronal cells [84,85].

We used MN cultures and in vivo sensors to monitor and detect whether any changes in metabolism could be detected in cells upon prolonged culture. We found that using microfluidic chambers not only allows the separation of compartments but also protects cultured MN from extra stress during media changes, increasing their survival time in comparison to larger cell formats. We have succeeded in robustly maintaining MN cells for more than 105 days by expressing the A-team sensor and continuously measuring lifetime values both in the soma and distal axons. Both our control lines showed a gradual increase in T_m_ values of A-team in the soma, reaching a statistically significant reduction in relative ATP levels by day 84, where it plateaued for the rest of the time course (Figure 5A–C), with T_m_(wt1) (day 84) = 1514 +/− 52.56, n = 17, *p* = 0.0003 and T_m_(wt2) (day 84) = 1363 +/− 31.71, n = 22, *p* = 0.047). Axons, although starting from lower ATP levels at baseline, also showed a significant increase in sensor lifetimes by day 84 (Figure 5B,E) (T_m_(wt1) (day 84) = 1514 +/− 52.56, n = 17, *p* = 0.0003 and T_m_(wt2) (day 84) = 1363 +/− 31.71, n = 22, *p* = 0.047), suggesting that FRET efficiency and thus relative ATP levels are indeed reduced. Therefore, metabolic rates slow down across all regions of neuronal cells after prolonged incubation in culture. Although 105 days of growth of cells in culture unlikely corresponds to real aging, prolonged culturing of cells in vitro has been extremely useful to uncover some aspects of neurodegeneration mechanisms. It has been suggested that growing cells in vitro may put them under higher levels of stress and can mimic at least some aspects of ‘aging’ in cells [23].

### 3.8. FUS ALS Mutations Lead to Lower ATP Levels in Somas of Cultured Motoneurons

Having established a more sensitive and in vivo way of measuring ATP levels, we then decided to re-visit ATP level measurements between control lines and three different FUSALS mutants available to us: two carrying mutations in the nuclear-localization signal (NLS) (FUS^R521C^, FUS^R495QsfX527^) and one with a mutation in the low complexity domain (LCD) (FUS^Q23L^). In support of previously suggested metabolism-related phenotypes, we observed lower relative ATP concentrations (higher lifetime values of the sensor) in the somas of all FUS mutant motoneurons at day 21 of neuronal maturation (Figure 6, Appendix A) with T_m_(FUS^R521C^) = 1383 +/− 30.15, n = 55, *p*= 0.0312, T_m_(FUS^Q23L^) =1413 +/− 43.27, n = 31, *p*= 0.0169), T_m_(R495QsfX527) =1443 +/− 32.34, n = 13, *p* = 0.0073). Later timepoint measurements of FUS-ALS axons were not possible due to the continuously developing structural degeneration in FUS-ALS axons after day 50 [23]. Therefore, our in vivo single-cell measurements detect an early relative reduction in ATP in the soma of all tested FUS-ALS mutant motoneurons.

### 3.9. FUS MN Have Reduced Viability upon Further Metabolic Challenge

Lower levels of ATP at such an early stage could facilitate the accumulation of potential cellular defects associated with incorrect FUS function, affect its phase-transition mechanisms, and overall make FUS-ALS MNs less resilient to any additional external stress. To address the above hypotheses, we wanted to assess if FUS-ALS MN are more vulnerable if challenged with additional metabolic stressors. We used MTT cell viability assays and treated three control and three FUS-ALS MN cell lines with well-characterized mitochondrial inhibitors: CCCP and Rotenone. Strikingly, two of the mitochondrial inhibitors at the presented concentrations did not significantly affect the viability of control cell lines but did clearly increase cell death in MN carrying FUS ALS mutations (Figure 7A, Appendix A). 

We finally measured the effect of glycolysis on cell survival. For this, we did the same tests together with either a block of glycolysis using 2-deoxy-d-glucose (2DG) or by inducing a shift of metabolism to glycolysis using glutamate [86,87,88,89]. Interestingly, using glutamate slightly improves the baseline viability of FUS-ALS MN. While blocking glycolysis reduces the viability of control and mutant MN, it is argued that glycolysis is still essential for MN growth. Together, these observations strongly support the role of FUS in maintaining the metabolic states of MN via the regulation of mitochondrial oxidative phosphorylation and supporting their normal function, homeostasis, and survival.

### 3.10. FUS Mutations May Cause Proton Leakage in Mitochondrial Membrane

For bulk metabolic analysis of patient-specific MN, we conducted seahorse measurements. Specific inhibition of different enzymes of glycolysis and the respiratory chain was performed while measuring the extracellular acidification rate (ECAR) as well as the oxygen consumption rate (OCR). An example graph is shown in Figure 7C. With the injection of port A (10 mM glucose, 1 mM pyruvate), the glycolytic increase over the baseline was measured, but there was no difference observed between wt and FUS-mutant MN. Injection of a complex V inhibitor in port B (1 µM oligomycin A) allowed us to probe the cells for proton leakage from the intermembrane space to the matrix by measuring the OCR. There was a significantly higher leakage observed in FUS mutant neurons when compared to *wt*. A decoupling agent was present in port C (0.7 µM FCCP). After its application, the resulting OCR is indicative of the maximum respiratory capacity. However, there was no difference observed when wild-type and diseased neurons were compared. Lastly, injection of port D (1.5 µM rotenone, 2.5 µM antimycin A, and 50 mM 2-DG) was used to block glycolysis in complex I and complex II to measure the absolute baseline of ECAR and OCR. ECAR and OCR measured during that period occur because of non-glycolytic acidification (e.g., Krebs cycle or breakdown of intracellular glycogen) and non-respiratory oxygen consumption (e.g., microsomal activities or cell surface oxygen consumption). We observed no differences between wt and FUS-mutant MN. Taken together, seahorses showed no global phenotypes in oxidative phosphorylation or glycolysis. However, it indicated a potential disturbance of the cristae structure by showing increased proton leakage after oligomycin A application.

## 4. Discussion

To understand how to prevent, treat, or even diagnose early such a severe and fast-progressing neurodegenerative disease as ALS, we need to understand and monitor changes that occur in vulnerable cells in the early disease course, even presymptomatic, and in the context of development and aging. ALS, and particularly FUS-ALS, is caused by a disruption of the complex set of molecular mechanisms in motoneurons and their surrounding cells [2,3,4]. In the majority of cases, however, changes occur incrementally, and these mechanisms become prominent and critical only later in life [70,85]. Aging itself is an equally complex and multifactorial process, but it is unambiguously accompanied by a slowing down of metabolism and, with it, disruption of protein-quality control systems [85]. Interestingly, these interlinked processes are especially critical for the etiology of many neurodegenerative diseases, not just FUS-ALS. 

Abundant direct and indirect evidence points out that metabolism does lie at the core of neurodegeneration and ALS [8,9,10,11,12,90,91]. Indeed, studies on at least FUS-ALS have reported defects in mitochondrial motility, morphology, and function and would suggest a consequent reduction in the efficiency of oxidative phosphorylation and metabolic stress for the highly energy-demanding motoneurons with the particular vulnerability of their axons [6,11,13,14,15,16]. However, it is not trivial to monitor and detect specific metabolic changes in neurons over time. At least one recent report showed that metabolic changes do occur during MN differentiation, but no differences in glycolytic or oxidative phosphorylation could be detected between control or FUS mutant motoneurons [10]. These observations are somewhat surprising given previously reported phenotypes, including mitochondrial depolarization, reduced motility, and fragmentation [23]. On the other hand, these results may reflect an inherent problem facing research on most neurodegenerative diseases, i.e., that initial critical changes at the early stages of the disease are subtle and incremental. Furthermore, these changes initially occur only in a specific neuronal sub-compartment. The reported mitochondrial defects manifest themselves first only in distal axons and are not initially visible in soma [7,13,82]. Keeping the above considerations in mind, we used previously established hiPSC-derived MN culture protocols and a range of techniques to re-visit measurements of metabolic rates in motoneurons. We aimed to (1) increase the sensitivity of our measurements; (2) assess changes in specific cell populations (namely neurons) and compartments of neurons, such as soma and axons; and (3) follow these measurements during differentiation, maturation, and after prolonged culturing of the MN cells to detect early changes and monitor their progression. 

Analysis of critical mitochondrial mRNAs, mitochondrial protein levels, high AEC ratios, NAD^+^/NADH concentrations, and redox ratios all clearly showed a shift towards higher metabolic rates and a shift towards OXPHOS in all MN upon differentiation and maturation, which supports expectations for the high energy demand of these highly specialized neuronal cells. FUS-ALS mutations did not affect most of these measurements and suggest that MN differentiation and growth occur normally in these cultures. We did, however, find the first indication that FUS-ALS MN have changes in their metabolism. We observed that global NAD^+^ concentrations were increased in all of the tested mutant lines. Although these changes did not significantly affect the NAD^+^/NADH redox ratio, often used to estimate OXPHOS rates in cells, they showed that critical metabolite concentrations are affected. Maybe there is a lack of mitochondrial conversion of NADH to NAD^+^ in FUS-ALS MN, which motivated us to use more precise, live, and single-cell imaging tools to asses metabolism in single MN and their subcompartments.

Utilization of FRET sensors coupled with FLIM imaging to measure ATP levels showed a much more detailed and nuanced picture in both control and FUS-ALS MN. Firstly, single-cell FLIM measurements highlight clear differences in ATP levels across cellular compartments in live cultured motoneurons. The somatic part of the cell has significantly higher ATP levels, which may reflect the abundance of mitochondria and all accessory machinery in this compartment. Distal axons, on the other hand, from early on, have significantly lower concentrations of ATP. These observations can represent a reduction in mitochondrial number/density due to physical distance from the cell body and/or potential changes in the metabolic pathways utilized in axons. Recent data have shown mitochondria anchoring at NMJs and highlighted their high energy demands [7]. Although our cultures did not contain NMJs, our measurements do emphasize that distance from the cell body or its compartmental metabolic specificity is a factor affecting energy availability in axons and puts them at a potentially higher risk of any further disruption of mitochondria and metabolism. The latter could explain why the deterioration of NMJ and distal axons and muscle denervation is one of the first manifestations of ALS in humans and in multiple studies of ALS model systems [7,13,82]. Alternatively, distal axons could be presented with a different set of metabolites, either due to distance from the cell body soma, mode of metabolism in axons, and/or differences in surrounding tissues. The importance of glycolysis to supplement the energy supply in the synapse, as well as creatine metabolism and lactate supply from surrounding glia, has been reported to play an important role in axonal function and resistance to neurodegenerative processes [92,93,94,95]. All of these are, however, not part of the in vitro culture system presented here. Our data prompt further investigation of what distinguishes the axonal compartment and which metabolites and pathways are involved and affected. We would like to further increase the resolution of our measurements in both soma and axons and, in the future, monitor mitochondria in parallel to our FLIM measurements to obtain a better idea of ATP concentration heterogeneity in relation to the number of these organelles.

Most strikingly, single-cell FLIM measurements were able to detect a significant reduction in ATP levels in all three tested FUS-ALS mutant lines, including the most severe mutation of FUS R495QsfX527, known for very early onset and aggressive subtype of ALS [66,96]. Thus, metabolic changes in FUS-ALS do occur early in affected cells, supporting previous observations, and are likely to play an important detrimental role in ALS progression. Multiple mechanisms have been proposed for the FUS’ direct and indirect involvement in regulating metabolism, including aggregate accumulation of mitochondrial transcripts or direct binding to glycolytic enzymes and ATP synthase subunits [6,11,13,14,15,16]. Our data from cell viability assays and oxygen consumption rate measurements strongly argues that mutated FUS does affect the function of mitochondria, increases proton leakage in the mitochondrial inner membrane, and makes FUS-ALS MN much more sensitive to further inhibition of mitochondria and OXPHOS. This might also explain increased NAD^+^ levels by disturbing the mitochondrial conversion of NADH to NAD^+^. Whether these defects are due to reported FUS localization to mitochondria or direct binding to metabolic enzymes will need to be addressed in further studies. One further area of interest would be to characterize in much more detail changes in the efficiency of glycolytic and OXPHOS pathways and their interaction in MN-carrying ALS mutations. An increase in FUS-ALS MNs’ survival upon stimulation of glycolysis as well as a clear shift in metabolic rates in all cell cultures after their prolonged cultivation suggests that some compensatory upregulation of glycolytic pathways could be beneficial to ALS MNs, as was reported in fALS models for mutations in *SOD1* and *TDP43* genes [97].

Interestingly, it was proposed that ATP concentration can regulate phase-transition properties of FUS and therefore potentially all transcription, translation, and RNA metabolism processes regulated by such properties of FUS and similar proteins [38]. The fact that we detect lower energy levels in all FUS mutant model cell lines could be one of the factors responsible for the gradual accumulation of cell defects in disease cells. Cells can tolerate these changes initially but may become much more sensitive to any potential further stress after the downregulation of metabolism that occurs in the cell and is proposed to happen in aging organisms. Indeed, we and others observe much higher cell death rates, frequency of FUS aggregation, and other ALS-related phenotypes in cells only after an extended time of cell culture and/or upon further externally added stress to the cell [23,53,98,99]. Such properties of our FUS-ALS model cells would be in strong agreement with the modern understanding of the progression of neurodegenerative diseases and their strong connection to the underlying mechanism of aging and accumulating damage from external factors.

Our measurements cover only a very short timeframe, and we cannot distinguish whether the observed reduction in ATP levels was due to a gradual change in metabolism in these cells or the potentially high-stress environment of culturing neurons in 2D without their normal support network of cells. One would need to address both of these questions in further studies using metabolomic approaches or comparing OCR rates and ATP levels in in vivo model systems and comparing these values in young and old organisms. 

In this study, we did not specifically control for potential gender-specific variation in the metabolism of healthy and FUS-ALS MN, and this issue would still require future investigation. Since we used both isogenic and non-isogenic conditions with no obvious differences in phenotypes, such gender-specific effects seemed rather small.

To our knowledge, there are very few examples of measurements in cell culture or in vivo models that demonstrate compartment-specific changes in metabolic rates with time. MN cultures could therefore be a powerful tool to study energy metabolism evolution in prolonged and ‘aged’ cultures. In the future, we would like to use our developed methodology to address several outstanding questions. By combining FLIM and mitochondrial markers, we would like to obtain further insight into what drives the reduction in ATP concentrations in soma and axons, if it is a global or mitochondrial-specific change, and if it is related to mitochondrial numbers and morphology. Furthermore, other metabolic stress conditions, e.g., glucose and/or oxygen deprivation as well as starvation, should be investigated to further decipher the metabolic disadvantage FUS-ALS MNs carries. We have already started to investigate which metabolic pathways could be particularly critical for somatic and axonal compartments by doing compartmentalized transcriptomics approaches, and we would like to extend it to protein and metabolite analysis. Finally, we would like to test if our methodology could work as an early diagnostic tool in animal models of ALS and could then potentially be used to study metabolism-based treatments for this complex and so far nearly untreatable disease.

## 5. Conclusions

To be able to correctly diagnose and potentially develop treatments for ALS, we have to be able to monitor cellular changes in much more detail, ideally with live and sub-compartmental resolution under physiological conditions. The FLIM-based non-invasive live cell imaging techniques that we have used, could be extremely beneficial for these purposes, particularly when combined with the development of more advanced and physiological models to study ALS, such as the development of new sensors and expressing them in model organisms. Furthermore, our data confirm the important role of FUS in regulating mitochondrial and metabolic homeostasis in motoneurons. These findings strongly support the idea of developing potential new treatments that target energy metabolism, mitochondrial damage, and oxidative stress in ALS.

## Figures and Tables

**Figure 1 cells-12-01352-f001:**
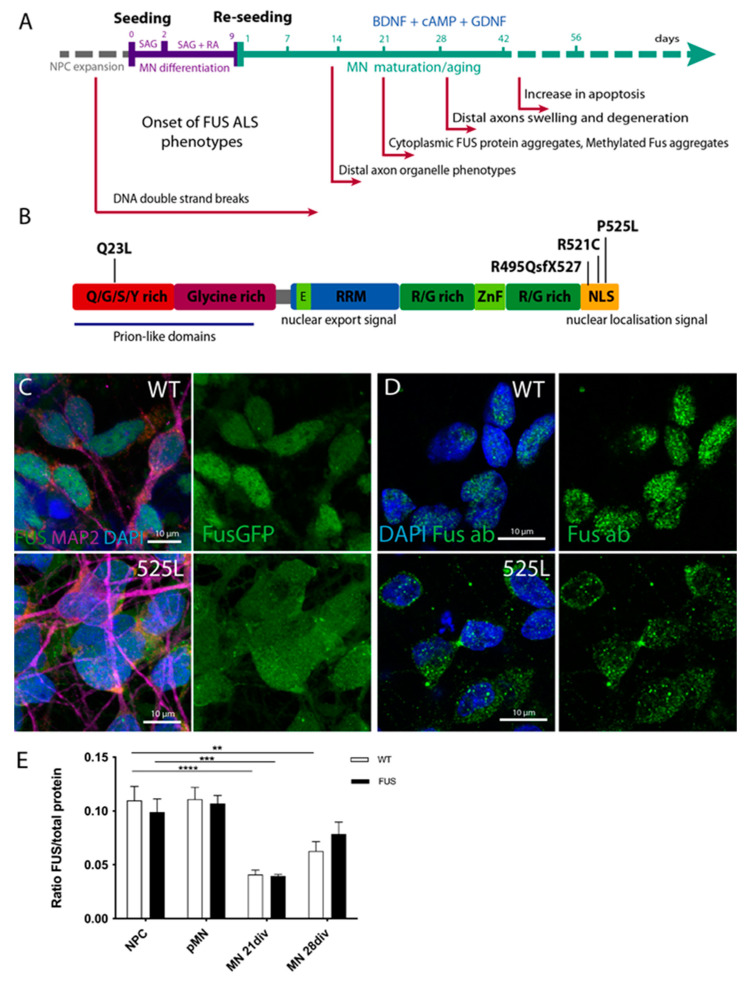
Generation of FUS-ALS MNs from human iPSCs. (**A**) Timeline and schematic outline of the motoneuron differentiation protocol starting from neural precursor cells and undergoing sequential differentiation and maturation. Time points for the onset of previously reported FUS-ALS-related phenotypes are indicated under the timeline. (**B**) Schematic drawing of known protein domains of FUS and locations of respective FUS mutations analyzed in this study. (**C**,**D**) Mature cultured MN showing mis-localization of mutant FUS^P525L^ protein from the nucleus to the cytoplasm, as seen by fluorescence imaging of FUS-GFP tagged isoforms (**C**) or by immunostaining with FUS antibody (**D**). (**E**) Bar chart quantification of Western blot analysis demonstrates the overall reduction in FUS protein levels during MN differentiation and maturation. Values were averaged from the individual measurements across all tested control and FUS mutant lines (see Appendix A for individual line evaluations). The *p* values in this and all subsequent figures are displayed as ** *p* < 0.01, *** *p* < 0.001, **** *p* < 0.0001. Scale bar: 10 µm. n ≥ 12 biological replicates.

**Figure 2 cells-12-01352-f002:**
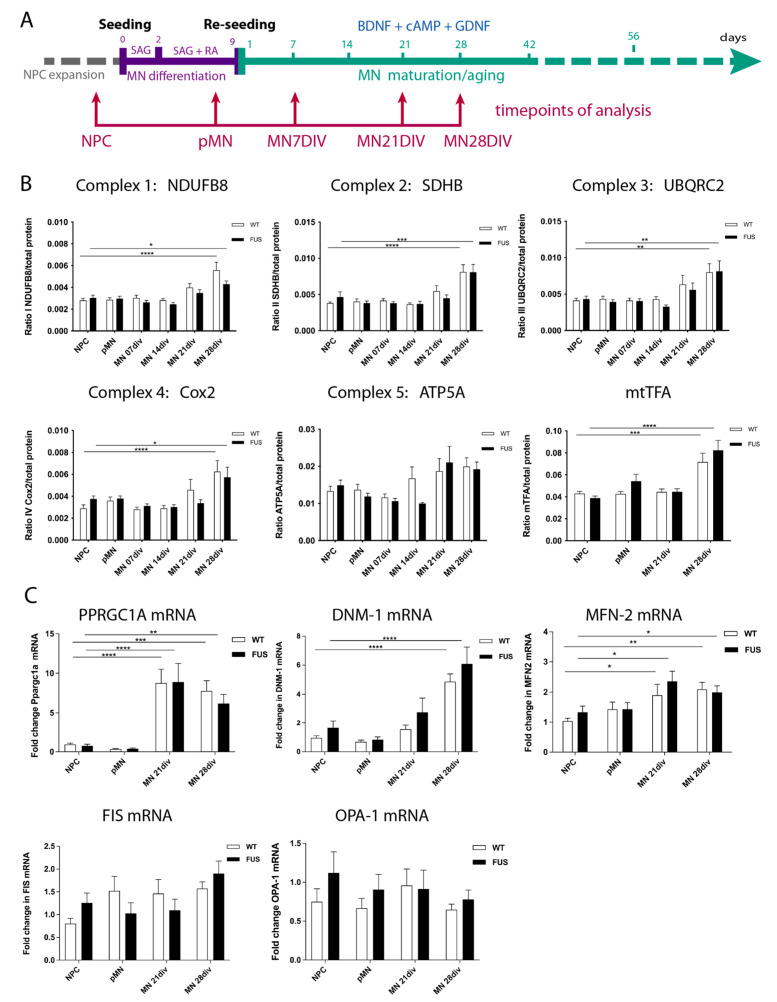
MN differentiation leads to an increase in mitochondrial complex proteins and mRNAs and is not affected by FUS. (**A**) Timeline and schematic outline of the motoneuron differentiation protocol showing timepoints at which cells were taken for analysis and are indicated by red arrows. (**B**) Western blot analysis for selected components of the mitochondrial respiratory chain subunits I-V: NDUFB8, SDHB, UQCRC2, COX2, ATP5A, as well as mitochondrial transcription factor A(mtTFA) shows an increase in the relative amounts of these proteins (excluding ATP5A) in 28-day-old MNs. (n ≥ 9 biological replicates (minimum three each line). (**C**) qPCR evaluation of several mRNAs critical for the maintenance and operation of the mitochondrial network, such as PPARGC-1α, Dnm1L, and MFN-2, shows an increase in the relative amounts by 21 or 28 days of MN maturation. Expression levels of OPA1 and FIS1 do not change. (n ≥ 15; five per line) biological replicates, n ≥ 2 technical replicas per sample. The presence of any FUS mutations did not significantly affect levels and amounts of any of the above-tested proteins and mRNAs (**A**–**C**). Values were averaged from the individual measurements across all tested control and FUS mutant lines (see Appendix A). Significance was tested by comparing cells at the NPC stage to all other timepoints. * *p* < 0.05, ** *p* < 0.01, *** *p* < 0.001, **** *p* < 0.0001.

**Figure 3 cells-12-01352-f003:**
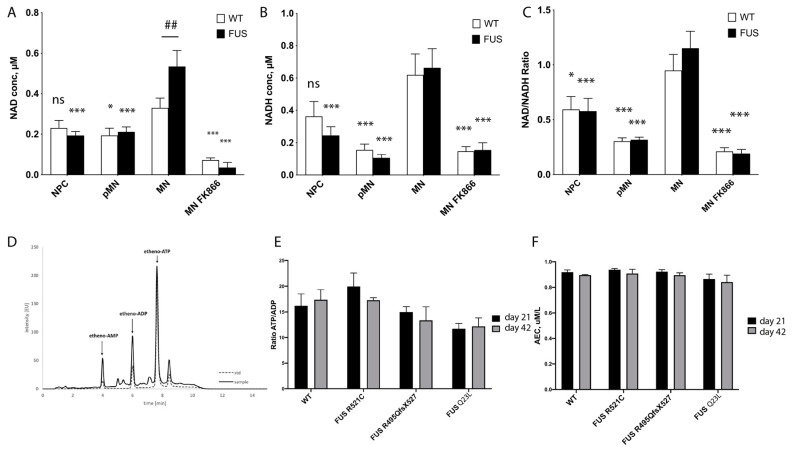
Evaluation of metabolic changes during MN differentiation of FUS-ALS cultured motoneurons. (**A**–**C**) Measurements of NAD^+^ (**A)**, NADH concentrations (**B**), and their redox ratios (**C**) were averaged across all tested WT and FUS MN lines and displayed as bar graphs. Significance testing comparing the values of mature MN to different time points of the same phenotype is shown with asterisks (*). Number signs (##) denote significance testing between WT and FUS cells within the same time point. Significance testing with two-way ANOVA with the Sidak method. For individual line measurements across all tested control and FUS mutant lines, see Appendix A. (n ≥ 13 biological replicates, * *p* < 0.05, *** *p* < 0.001). (**D**) An example of a typical chromatogram of adenine nucleotides in a whole-cell motoneuron extract sample. (**E**,**F**) Comparison of measured ATP/ADP concentration ratios (**E**) and adenylate energy charge (AEC) (**F**) in control (n = 20) and FUS-ALS mutants (R521C (n = 9), R495QsfX527 (n = 6), and Q23L (n = 6) at two different maturation time points: day 21 and day 42 of MN maturation. We did not detect significant changes in FUS ALS mutants at either time point.

**Figure 4 cells-12-01352-f004:**
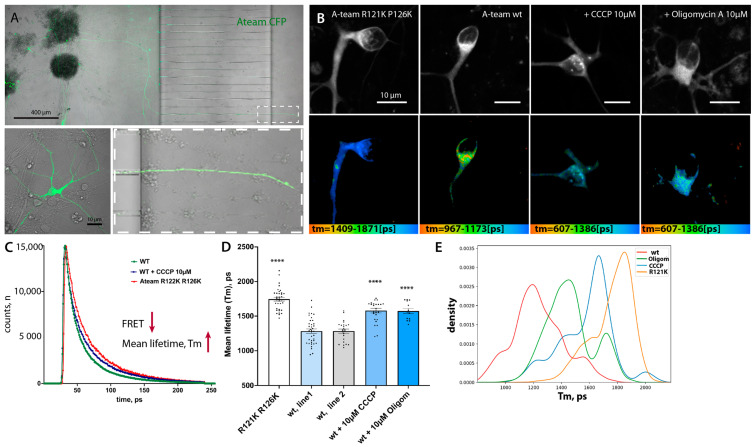
Lentiviral delivery of the FRET-based ATP sensor A-team allows metabolic measurement at a single-cell level and compartmental resolution. (**A**) Low magnification of the microfluidic chamber demonstrating motoneurons transfected with lentivirus. Low to moderate transfection efficiency allows the identification of single cells and measurements and tracing of individual soma, axonal, and distal axonal compartments within each chamber. (**B**) Representative single images showing the fluorescent intensity of the donor (upper panel) and the corresponding color-coded range of mean-lifetime FLIM measurements (lower panel) in transfected MN cells. (**C**) Example single-pixel FLIM decay model curves show a shift towards an increase in mean lifetime values in cells treated with the mitochondrial blocker CCCP or transfected with a non-FRET control sensor, A-team R121K R126K. (**D**) Bar-graph chart showing changes in mean lifetime measurements using A-team. Baseline FLIM measurements of the non-FRET donor fluorophore in a mutagenized A-team R121K R126K sensor (n = 36), comparable average lifetime measurements in two different untreated control cell lines (wt1, n = 36; wt2, n = 24), and a strong reduction in Tm values upon treatment with two different mitochondrial blockers, CCCP (10 uM) and oligomycin A (10 µM). The significance displayed is an unequal variance T-test comparing wt1 to its corresponding treatments (n_(CCCP)_ = 26, n_(Olig)_ = 13). **** *p* < 0.0001. (**E**) Kernel density estimate (KDE) plot showing shifts in normalized distributions of mean lifetime values per pixel in cells transfected with a non-FRET control sensor (R121K R126K) or with standard A-team without treatment (line wt1) or treated with two different mitochondrial blockers, CCCP (10 µM) and oligomycin A (10 µM).

**Figure 5 cells-12-01352-f005:**
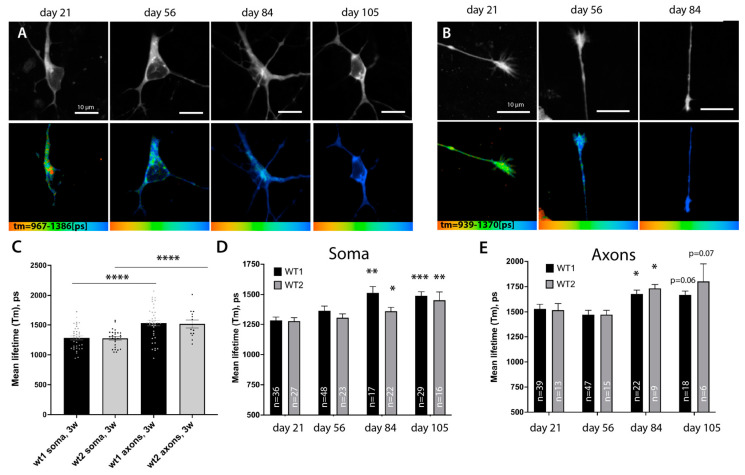
A-team FLIM revealed lower levels of ATP in distal axons and a global slowing down of metabolism rates during prolonged culture of motoneurons. (**A**,**B**) Representative images showing the fluorescent intensity of the donor (upper panel) and the corresponding color-coded range of mean-lifetime FLIM measurements using A-team (lower panel). Images show different cell compartments, soma (**A**) and axons (**B**), in control MN cells at different timepoints of cell culture: days 21, 56, 84, and 105 of maturation and aging. (**C**) Bar graphs comparing average mean lifetimes in the soma (n_wt1_ = 36, n_wt2_ = 27) and axons (n_wt1_ = 39, n_wt2_ = 13) of two independent control lines of MN show a strong and significant reduction in relative ATP levels in distal axonal compartments at day 21. (**D**,**E**) Prolonged culturing of MN leads to a significant global reduction in supposed ATP concentrations across neurons. Bar chart showing an increase in FLIM lifetimes of A-team donor fluorophore at various time-points post differentiation in soma (**D**) and distal axon (**E**) in two independent control lines upon long-term growth in culture. In D and E, an unequal variance T-test was used to compare the values between the first timepoint 21 and later timepoints of the same line. * *p* < 0.05, ** *p* < 0.01, *** *p* < 0.001, **** *p* < 0.0001.

**Figure 6 cells-12-01352-f006:**
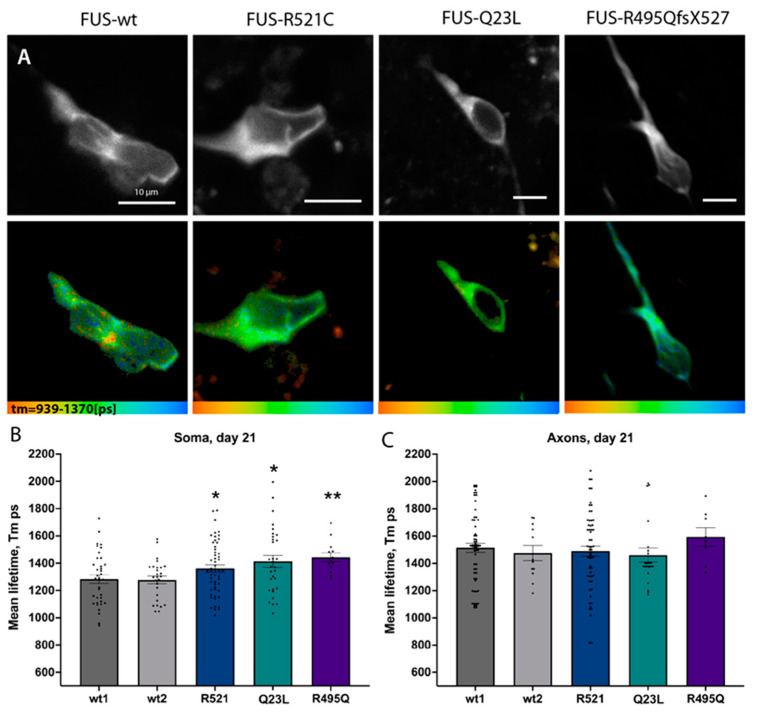
FUS ALS mutations led to early changes in ATP levels in cultured motoneurons. (**A**) Representative single images showing the fluorescent intensity of the donor (upper panel) and the corresponding color-coded range of the mean-lifetime from FLIM measurements (lower panel) in 21-day-old mature control or FUS R521C, Q23L, and R495QfsX527 mutant MNs. (**B**,**C**) Bar graphs comparing average mean lifetimes in the soma (**B**) and axons (**C**). Significances display differences between the control line (wt1) and mutants and were assessed with an unequal variance unpaired *t*-test. * *p* < 0.05, ** *p* < 0.01.

**Figure 7 cells-12-01352-f007:**
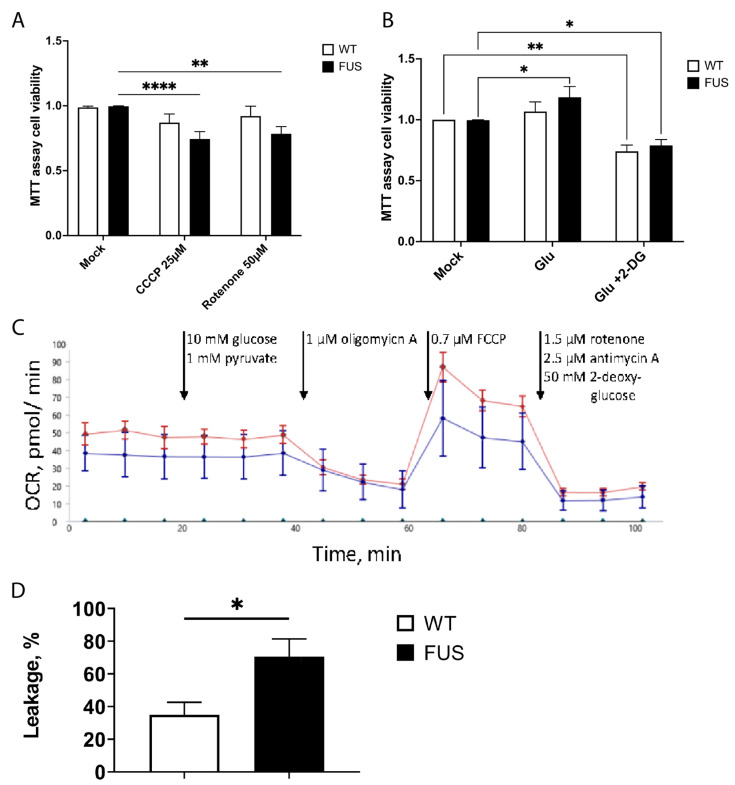
FUS ALS motoneurons have reduced viability upon additional mitochondrial stress and have increased proton leakage. (**A**) Bar graphs comparing cell survival of control and FUS ALS mutant MNs using MTT cell viability assays treated with mitochondrial inhibitors CCCP and rotenone. Both mitochondrial inhibitors only affected FUS mutant MN. ** *p* < 0.01, **** *p* < 0.0001, two-way ANOVA (Dunnet’s multiple comparison test) (n ≥ 17 biological replicates, n = 2 technical replicas). (**B**) Glutamate stimulation of motoneurons and switching of metabolism to glycolytic pathways slightly improve the baseline viability of FUS-ALS MN, while blocking glycolysis by the utilization of 2-deoxy-d-glucose (2DG) reduces the viability of both wt and FUS mutant MN. n ≥ 26 biological replicates per phenotype, n = 2 technical replicas. * *p* < 0.05, ** *p* < 0.01. (**C**) Example graph of seahorse measurement, including the injections during the assay (arrows). (**D**) FUS mutant neurons have increased proton leakage through the inner mitochondrial membrane. This result is calculated from OCR before and after oligomycin A injection. Bars represent mean ± SEM; * represents *p* < 0.05; n > 7 biological replicates each; N = 6 technical replicates.

**Table 1 cells-12-01352-t001:** Cell lines were used in this study.

	FUS Line	Sex	Age at Biopsy	Mutation	Primarily Characterized in
Wt	Ctrl1	Female	43	-	[52]
Wt	Ctrl2	Female	49	-	[52]
Wt	Ctrl3	Female	48	-	[53]
Wt	Ctrl4-GFP	-	-	-	[23]
Mt	FUS-1	Female	58	R521C	[53]
Mt	FUS-2	Female	65	R521L	[23]
Mt	FUS-3-GFP	-	-	P525L	[23]
Mt Mt	FUS-4 FUS-5	Male Male	29 46	R495QfsX527 Q23L	[23,53] This study

## Data Availability

All data are presented in the manuscript.

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
