# Peer review of "Live Cell Imaging of ATP Levels Reveals Metabolic Compartmentalization within Motoneurons and Early Metabolic Changes in FUS ALS Motoneurons"

_cells, 2023, doi:10.3390/cells12101352_

Round 1

Reviewer 1 Report

In the present paper, Zimyanin and collaborators utilized iPSC-derived human motor neuron cultures to investigate changes in mitochondrial metabolism and ATP levels in different subcellular compartments and to unveil potential metabolic defects in cells obtained from ALS patients carrying FUS mutations compared to healthy subjects. The research is timely and address a key aspect of ALS research, namely early pathological alterations caused by bioenergetic dysfunction. Overall, this study has been carried out with solid and rigorous methodology. Nonetheless, relatively easy experiments that are missing may be added to significantly enrich the impact of the results presented.

Specific comments are reported below:

1) Measuring protein/mRNA levels of mitochondrial markers gives little information on the actual morphology and dynamics of mitochondria. Adding analysis of mitochondrial morphology by EM and dynamics using live imaging and fluorescent mitotrackers may add invaluable information about possible mitochondrial dysfunction in FUS motor neurons.

2) The results emerging when culturing WT cells for very long time (up to 105 days) are interesting. I strongly encourage including similar data or discussion also for FUS cells (is there a viability issue?), which may unmask stronger defects compared to those observed at 21 days.

3) Also the data regarding metabolic challenge experiments are intriguing. What happens if, instead of blocking metabolic pathways with specific artificial inhibitors, the cells are grown under more physiological starvation, i.e. glucose or oxygen deprivation? This condition may resemble in a closer way the metabolically stressful environment that motor neurons encounter during disease progression, providing more translatable results.

4) Please make explicit in the figure legend the number of biological and technical replicates for each analysis.

Author Response

Dear Editor, dear reviewers,

we are very grateful for the overall very positive feedback for our manuscript and very much appreciated the remaining constructive suggestions raised by the three reviewers. As you can find below, we tried hard to adress and answer every single comment. We believe that this significantly strengthens our manuscript and hope that it is now acceptable for publication.

Comments and Suggestions for Authors

In the present paper, Zimyanin and collaborators utilized iPSC-derived human motor neuron cultures to investigate changes in mitochondrial metabolism and ATP levels in different subcellular compartments and to unveil potential metabolic defects in cells obtained from ALS patients carrying FUS mutations compared to healthy subjects. The research is timely and address a key aspect of ALS research, namely early pathological alterations caused by bioenergetic dysfunction. Overall, this study has been carried out with solid and rigorous methodology. Nonetheless, relatively easy experiments that are missing may be added to significantly enrich the impact of the results presented.

Response: We deeply thank for the very positive review.

Specific comments are reported below:

1) Measuring protein/mRNA levels of mitochondrial markers gives little information on the actual morphology and dynamics of mitochondria. Adding analysis of mitochondrial morphology by EM and dynamics using live imaging and fluorescent mitotrackers may add invaluable information about possible mitochondrial dysfunction in FUS motor neurons.

Response: We extremely value reviewer comment since similar kind of analysis was exactly our basis to begin measurements we describe in this manuscript. The lines we have described in this study have have previously charachterised in detail (PMID: 30422121 [60]; PMID: 29362359 [23]). Large part of these published work was extensive light microscopy analysis of the morphology and dynamics of organelles with special focus on mitochondria. In Naumann et al, for example (which we reference [23]) we compared the behaviour and morphology of mitochondria and lysosomes morphology and motility in somatic, as well as proximal and distal axonal compartments. We could observe that although DNA damage was the earliest detectable phenotype, with culturing time it leads to a disruption in motility, morphology and depolarisation of mitochondria in the distal axon, but not in the proximal axon or soma. Notably this is first detected in the most distal axonal compartment and could be one of the reasons why distal axonal compartments start deteriorating first. Several other studies have demonstrated mitochondrial defects in FUS MNs as well [6, 7, 11, 13, 17, 25]. These and other results motivated our cellullar and compartment specific mesurements of metabolism that we describe here.

Thus, knowing about the morphological phenotypes in FUS-ALS neurons, the analysis of protein/mRNA levels of mitochondrial markers was the first step to investigate how differences in the metabolic rates and morphological phenotypes might be explained, more or less as ground for the later measurements in the paper.

Nevertheless we are adding a supplementary figure, where we present data we obtained using mitochondrial TMRE dye. Using this dye we could validate our and other published results the FUS-ALS mutations lead to a disruption of mitochondrial membrane potential, and this disruption is most pronounced in distal axonal compartment.

Electron microscopy is also a very exciting proposal and we tried this in the past a couple of times. Although EM, may not be able to detect membrane depolarisation, it would extremely exciting to look at the changes in morphology and structure of mitochondria of FUS-ALS MN at a significantly higher resolution. However we yet failed to be able to do it specifically in distal axons.

2) The results emerging when culturing WT cells for very long time (up to 105 days) are interesting. I strongly encourage including similar data or discussion also for FUS cells (is there a viability issue?), which may unmask stronger defects compared to those observed at 21 days.

Response: In this study we did not yet do this exact proposed experiment. We have known from our previous research that FUS-ALS lines start showing signs of axonal swellings, dramatic damage and shrinking back of axons after DIV30 ( very obvious after DIV60, please refer to Fig 2 of Naumann et al, PMID: 29362359). Thus it is just impossible to measure this at later timepoints. We made this more clear in the revised version.

3) Also the data regarding metabolic challenge experiments are intriguing. What happens if, instead of blocking metabolic pathways with specific artificial inhibitors, the cells are grown under more physiological starvation, i.e. glucose or oxygen deprivation? This condition may resemble in a closer way the metabolically stressful environment that motor neurons encounter during disease progression, providing more translatable results.

Response: We thank reviewer for this comment. This was partially our rationale and ideas why we started to add Glycolysis inhibitors in our measurements. Unfortunately we did not have done these other experiments. We added this in the end of the discussion as limitations/ future outlook in the revised version.

4) Please make explicit in the figure legend the number of biological and technical replicates for each analysis.

Response: We went back through our figures and added the requested numbers either in all figure legends, text or in actual figures. We apologise for not noticing this omisions. We also added a sentence in methods part that all of experiments had minimum 3 biological replicates.

Reviewer 2 Report

The article proposed by Zimyanin and colleagues is well-structured and has a good scientific impact. The text is well-written despite small errors, such as on lines 136 and 146.
The biggest shortcoming of the study is the lack of a cell line of healthy male volunteers: all the controls are from females or generated with Crisp/Cas mutagenesis. This lack must be implemented.

Author Response

Dear Editor, dear reviewers,

we are very grateful for the overall very positive feedback for our manuscript and very much appreciated the remaining constructive suggestions raised by the three reviewers. As you can find below, we tried hard to adress and answer every single comment. We believe that this significantly strengthens our manuscript and hope that it is now acceptable for publication.

Comments and Suggestions for Authors

The article proposed by Zimyanin and colleagues is well-structured and has a good scientific impact. The text is well-written despite small errors, such as on lines 136 and 146.

Response: We deeply thank for this very positive review and went thorougly through the manuscript to remove these errors.

The biggest shortcoming of the study is the lack of a cell line of healthy male volunteers: all the controls are from females or generated with Crisp/Cas mutagenesis. This lack must be implemented.

Response: We totally agree with the reviewer on this point, but have to say that it accidentally happened to not being perfectly matched. We appologise for this but cannot complement this quickly since we would have to re-perform the studies with 1 additional control line.

Of note, however, are two facts:

  1. We have roughly 50-50% split in the gender of patients for our mutant lines and we do not observe any strong consistent gender-specific variation within measurements of the same phenotype.
  2. In addition, many of our measurements contain an isogenic pair that is different only by a single point mutation in FUS and results from this controlled pair are consistent with all other measurements. In this case we can argue that observed changes would be driven only by the modification in the function of FUS protein.

This gives us hope that overall our metabolic measurements remain valid and equally characteristic for the FUS-ALS cell lines independent of their gender of origin.

We do acknowledge that gender differences can seriously affect measurements and that we cannot completely rule this possibility out. We thus added this as putative caveat in the limitations section of the revised version.

Reviewer 3 Report

The current article by Zimyanin et al. is an interesting piece of work where the authors have used live cell imaging technique for to evaluate metabolic rates in Fused in Sarcoma (FUS)-ALS model cells. This can be accepted for publication, however, I have some queries.

The introduction section is too length. Please narrow it down. 

Line 390-391: It can be a part of discussion section.

Line 485 to 491: Please move it to discussion section. 

The authors should focus on explaining the results in the Results section. Authors have already provided a lot of background information in the introduction part so there is no need to provide the background in the Results section. Please either get rid of it or move it to the discussion section.

What are the future prospects of current study. Please also include it at the end of conclusion section.

Author Response

Dear Editor, dear reviewers,

we are very grateful for the overall very positive feedback for our manuscript and very much appreciated the remaining constructive suggestions raised by the three reviewers. As you can find below, we tried hard to adress and answer every single comment. We believe that this significantly strengthens our manuscript and hope that it is now acceptable for publication.

Comments and Suggestions for Authors

The current article by Zimyanin et al. is an interesting piece of work where the authors have used live cell imaging technique for to evaluate metabolic rates in Fused in Sarcoma (FUS)-ALS model cells. This can be accepted for publication, however, I have some queries.

Response: We deeply appreciate the reivewer’s overall remark and provisionally accepting our manuscript.

The introduction section is too length. Please narrow it down. 

Response: We have shortened the introduction in the revised manuscript.

Line 390-391: It can be a part of discussion section.

Response: We moved this information to the discussion.

Line 485 to 491: Please move it to discussion section. 

Response: This sentences were moved and were reduced to a single introductury sentence.

The authors should focus on explaining the results in the Results section. Authors have already provided a lot of background information in the introduction part so there is no need to provide the background in the Results section. Please either get rid of it or move it to the discussion section.

Response: We have significantly shortened introductions in the sections 3.4 and 3.5, 3.6 3.7

What are the future prospects of current study. Please also include it at the end of conclusion section.

Response: We have added our ideas for the future prospects and some of our further already ongoing experiments just before our conclusions section.

Round 2

Reviewer 1 Report

The authors comprehensively addressed all the comments of this Reviewer.